# Differential Regulation of Hemichannels and Gap Junction Channels by RhoA GTPase and Actin Cytoskeleton: A Comparative Analysis of Cx43 and Cx26

**DOI:** 10.3390/ijms25137246

**Published:** 2024-06-30

**Authors:** Oscar Jara, Jaime Maripillán, Fanny Momboisse, Ana María Cárdenas, Isaac E. García, Agustín D. Martínez

**Affiliations:** 1Instituto de Neurociencias, Centro Interdisciplinario de Neurociencia, Universidad de Valparaíso, Valparaíso 2362807, Chile; 2Department of Pediatrics, University of Chicago, Chicago, IL 60637, USA; 3Virus and Immunity Unit, Institut Pasteur, Université Paris Cité, CNRS UMR3569, 75013 Paris, France; 4Laboratorio de Fisiología Molecular y Biofísica, Facultad de Odontología, Universidad de Valparaíso, Valparaíso 2360004, Chile; 5Centro de Investigación en Ciencias Odontológicas y Médicas, CICOM, Universidad de Valparaíso, Valparaíso 2360004, Chile

**Keywords:** connexin trafficking, cytochalasin-B, small GTPase, hemichannels, gap junction channel regulation

## Abstract

Connexins (Cxs) are transmembrane proteins that assemble into gap junction channels (GJCs) and hemichannels (HCs). Previous researches support the involvement of Rho GTPases and actin microfilaments in the trafficking of Cxs, formation of GJCs plaques, and regulation of channel activity. Nonetheless, it remains uncertain whether distinct types of Cxs HCs and GJCs respond differently to Rho GTPases or changes in actin polymerization/depolymerization dynamics. Our investigation revealed that inhibiting RhoA, a small GTPase that controls actin polymerization, or disrupting actin microfilaments with cytochalasin B (Cyto-B), resulted in reduced GJCs plaque size at appositional membranes and increased transport of HCs to non-appositional plasma membrane regions. Notably, these effects were consistent across different Cx types, since Cx26 and Cx43 exhibited similar responses, despite having distinct trafficking routes to the plasma membrane. Functional assessments showed that RhoA inhibition and actin depolymerization decreased the activity of Cx43 GJCs while significantly increasing HC activity. However, the functional status of GJCs and HCs composed of Cx26 remained unaffected. These results support the hypothesis that RhoA, through its control of the actin cytoskeleton, facilitates the transport of HCs to appositional cell membranes for GJCs formation while simultaneously limiting the positioning of free HCs at non-appositional cell membranes, independently of Cx type. This dynamic regulation promotes intercellular communications and reduces non-selective plasma membrane permeability through a Cx-type dependent mechanism, whereby the activity of Cx43 HCs and GJCs are differentially affected but Cx26 channels remain unchanged.

## 1. Introduction

Connexins (Cxs) are proteins widely expressed in many animals’ tissues [1,2]. Six Cxs subunits oligomerize to form hemichannels (HCs) inserted into the plasma membrane to connect the extra- and intracellular milieu. The docking of two HCs, each provided by neighboring cells, forms intercellular aqueous pores named gap junction channels (GJCs), permeable to ions and diverse metabolites [2,3]. There are 21 and 20 Cxs genes in humans and mice, respectively, and cells can express more than one Cx [4]. Cx oligomerization and transport to the plasma membrane can use different pathways, depending on the isoform. For example, Cx32 oligomerizes in the Endoplasmic Reticulum-Golgi Apparatus (ER-Golgi) and then is transported to the plasma membrane, whereas Cx26 is transported to the plasma membrane, bypassing the Golgi apparatus [5]. On the other hand, Cx43 HCs oligomerizes in the Trans-Golgi Network (TGN), and treatment with brefeldin A prevents the traffic of Cx43 HCs to the plasma membrane [6]. These results suggest that Cx32 and Cx43 reach the plasma membrane following the classic secretory pathway and that Cx26 follows an alternative route [7,8]. This alternative trafficking mechanism may constitute a way for faster channel insertion into the plasma membrane [9]. However, some studies have suggested that a proportion of Cx26 travels to the plasma membrane using the canonical TGN-dependent secretory pathway [10,11]. The cytoskeleton plays a significant role in the trafficking of Cxs; nevertheless, it seems that its mechanisms could be different depending on the cytoskeletal protein or the type of Cx involved. For example, actin depolymerization affects the assembly of GJCs formed by Cx26 or Cx43, but only Cx43 is sensitive to the disruption of microtubules [12]. Evidence shows that Cx43 connexons form GJCs through a microtubule-dependent pathway that requires the association with the microtubule plus-end-tracking protein (EB1) and its interacting protein, p150 [13]. However, whether or not the cytoskeleton regulates the trafficking and function of Cx26 is controversial, since some studies suggest that Cx26 utilizes a microtubule-dependent pathway to the plasma membrane [7,9,14] and others discard this possibility [12]. Similarly, some studies suggest that Cx26 has actin-dependent [11] or independent [14] mechanisms to form GJCs.

Actin is one of the most abundant intracellular proteins. The G-actin (globular monomer) and the F-actin (polymeric filament) are in constant and dynamic equilibrium. This process requires hydrolysis of adenosine triphosphate (ATP) following filament assembly, which allows the formation of a dynamic cytoskeleton. The actin cytoskeleton is involved in several intracellular processes that require synchronized and complex regulation by specific proteins, such as Rho GTPases [15,16,17,18]. Rho GTPases are critical for actin cytoskeleton rearrangements necessary for stress fibers formation, cell adhesion, and migration, processes that require fine temporal and unique tuning between actin polymerization and depolymerization [19,20]. Previous studies have shown the participation of RhoA in the regulation of Cx43 trafficking to the plasma membrane [21] or gap junction formation, since inhibition of Rho with C3 reduces the assembly of Cx43 gap junctions in corneal epithelial cells [22]. This type of regulation could be more general, in some cases resulting in different effects on Cx expression. For example, the activation of the RhoA/ROCK pathway in mononuclear cells from dialysis patients with atrial fibrillation showed an increased expression of Cx40 compared to the control group [23]. However, activation of the RhoA/ROCK pathway reduced the expression of Cx43 gap junctions in retinal endothelial cells in a mouse model of diabetic retinopathy [24]. In addition, Rho GTPases are essential regulators of lens morphology, since their inactivation induces profound lens abnormalities and cataracts in mice, which is partially mediated by a reduced expression of Cx50 [25]. However, since Cxs can make HCs and GJCs, we think it is essential to determine whether actin remodeling and RhoA GTPase signaling affect both channels similarly and whether this regulatory process is dependent on the Cx isoform. Therefore, in this study, we evaluated the role of actin microfilament dynamics and RhoA activity on the trafficking and functional state of HCs or GJCs made by Cx43 and Cx26, which are two Cxs that present marked differences in trafficking and functional regulation.

## 2. Results

### 2.1. Inhibition of RhoA Reduces the Size of Cx43 and Cx26 GJCs Plaques and the Functional State of GJCs Formed by Cx43 but Not by Cx26

To determine the role of RhoA in stress fiber and GJCs formation, we first incubate HeLa cells expressing Cx43 or Cx26 with 0.5 μg/mL of the C3 exoenzyme from clostridium botulinum, which is a non-selective inhibitor of Rho GTPases covalently linked to a cell-penetrating moiety. After treatment, cells were immunolabeled with anti-Cx43 or anti-Cx26 and stained with phalloidin to visualize the actin filaments (Figure 1a, upper and bottom panel). Quantification of stress fibers and GJCs plaque size shows a similar decrease in the number of stress fibers and GJCs plaque size formed by either Cx43 or Cx26 (Figure 1b–e).

To further address whether RhoA modulates stress fibers and GJCs plaque size in cells, we transfected cells with GFP-tagged specific cDNA constructs: (1) siRNAs to reduce the expression of RhoA, (2) a dominant negative form of RhoA (RhoAN19) to inhibit its function, and (3) a wild-type form of RhoA (RhoA-WT) or the constitutively active form of RhoA (RhoAV14) [20,23] to overexpress RhoA in cells. We found that expression of RhoAN19 produced a 75 and 66.3% reduction in the size of GJCs plaques formed by Cx43 and Cx26, respectively (*p* < 0.05, *n* = 20 cell pairs). Similarly, we found a 16% reduction in the size of GJCs plaques formed by Cx43 and a 41.4% reduction in GJCs plaques formed by Cx26 in cells transfected with siRNA RhoA (*p* < 0.05, *n* = 20 cell pairs). As expected, there were no differences in the size of GJCs plaques in cells transfected with the empty vectors or the RhoAWT, RhoAV14, or siRNAUnR constructs used as controls (Appendix A).

Next, we performed dye coupling experiments. We validate our system using HeLa Parental, HeLa-Cx43, and HeLa-Cx26 cell lines. The HeLa Parental cells showed no dye transfer to the neighboring cells. In contrast, HeLa-Cx43 and HeLa-Cx26 exhibited dye transfer to the adjacent cells, which was blocked by carbenoxolone (CBX, 20 µM) in both cases (Appendix A). To evaluate the role of RhoA in the functional state of GJCs expressed in HeLa-Cx43 and HeLa-Cx26 cell lines, we treated both cell lines with 0.5 μg/mL of C3 for 0.5, 1, 2, and 3 h, and the functional state of the GJCs was evaluated by microinjection of Lucifer Yellow (LY) (Figure 2a,b). We found that incubation with C3 reduced dye transfer between cells expressing Cx43, reaching maximum reduction after three hours (Figure 2c). Interestingly, even though the size of the GJCs plaques formed by Cx26 was reduced in HeLa-Cx26 cells treated for 3 h with 0.5 μg/mL of C3 (Figure 1b), the coupling remained unaffected (Figure 2d).

### 2.2. Inhibition of RhoA Increases the Activity of HCs Formed by Cx43 but Not by Cx26

Cx43 and Cx26 HCs can be delivered to the appositional plasma membranes to form GJCs but these HCs may also reach non-appositional plasma membranes, where they can remain as free HCs that participate in paracrine and autocrine functions. HeLa cells expressing Cxs displayed functional HCs at the plasma membrane, as indicated by the fluorescence intensity resulting from ethidium uptake. In contrast, the HeLa Parental cell line, which does not express connexins, showed minimal fluorescence intensity, suggesting a lack of connexin-mediated ethidium entry into the cells (Appendix A). To investigate if the functional state of HCs formed by Cx43 or Cx26 is under the control of RhoA, we used C3 to inhibit RhoA and assessed the plasma membrane permeability to ethidium, as previously reported [26,27]. Cells were treated with 0.5 μg/mL of C3 for 0.5, 1, 2, and 3 h. We observed that the incubation with C3 produced a time-dependent increment in the ethidium uptake in cells expressing Cx43 (Figure 3a) but not in cells expressing Cx26 (Figure 3c). We bathed cells in a divalent cations-free solution (DCFS; without Ca^2+^/Mg^2+^) to promote HCs opening. Under this condition, a peak of 2.4-fold increments of ethidium uptake rate compared to control (*n* = 4; *p* < 0.05) was observed after 1 h in HeLa-Cx43 cells treated with C3 (Figure 3b). No changes in ethidium uptake rate were found in HeLa-Cx26 cells treated with C3 (Figure 3c). Similarly, HeLa-Cx43 cells transfected with siRNARhoA or with the dominant negative form of RhoA (RhoAN19) also present a high ethidium uptake rate (2.3-fold, *n* = 4, *p* < 0.05) compared to cells transfected with RhoAWT, RhoAV14, and the empty vector (Figure 4a,b). As expected, no differences in the rate of ethidium uptake were found in cells expressing Cx26 transfected with different RhoA constructs (Figure 4c,d).

### 2.3. Actin Depolymerization Mimics the Effect of Inhibition of RhoA on Cx43 and Cx26 GJCs and HCs

Since RhoA inhibition reduces the number of actin fibers (Figure 1d,e), we decided to evaluate the direct effect of actin depolymerization on the formation and functional state of Cx43 and Cx26 GJCs. Actin depolymerization was achieved by treating cells with cytochalasin B (Cyto-B, 4 μM) for 10, 30, and 60 min in HeLa-Cx43 (Figure 5a, upper panels) and HeLa-Cx26 (Figure 5a, bottom panels). Cyto-B treatments induced a time-dependent reduction in the size of GJCs plaques formed by both Cx43 (Figure 5b) and Cx26 (Figure 5c). Interestingly, Cyto-B treatment only reduced dye coupling between cells expressing Cx43 (Figure 6a,c), not in cells expressing Cx26 (Figure 6b,d).

Moreover, after one hour of exposure to Cyto-B (4 µM), cells expressing Cx43 showed a strong ethidium uptake rate (1.8-fold compared to control, *n* = 4, *p* < 0.001) in the DCFS milieu that was inhibited by the unspecific HCs blocker La^3+^ (Figure 7a,b). This effect was absent in cells expressing Cx26 (Figure 7c,d). As observed before in C3-treated cells, divalent cations abrogate the increase in ethidium uptake in cells expressing Cx43 or Cx26, suggesting that, as with RhoA inhibition, actin depolymerization by itself cannot induce the opening of Cx43 or Cx26 HCs.

### 2.4. Actin Depolymerization Steadily Increases the Transport of HCs to Non-Appositional Plasma Membranes

The opposite effect on GJCs and HCs activity induced by the inhibition of actin polymerization may result from changes in the destination of HCs to the plasma membrane, leading to a reduction in the number of GJCs but an increment in the number of functional HCs. Previous studies have shown that Cx43 connexons may traffic directly to appositional zones through tubulin-based transport to form GJCs [13,26,28]. We hypothesize that actin depolymerization disrupts this polarized trafficking of connexons to appositional zones, which results in increased transport to non-appositional zones to form free HCs available to be open by appropriate stimuli. To interrogate this possibility, we used total internal reflection fluorescence (TIRF) microscopy to simultaneously image HCs transport to appositional and non-appositional plasma membrane zones (Figure 8a). Besides the capability of identifying two cells expressing GJCs (Figure 8b, left panel), TIRF includes a surface reflection interference contrast (SRIC) filter to visualize the interface between the membrane and the glass where the cells are attached (Figure 8b, middle panel). HeLa-Cx43EGFP cells were treated with Cyto-B for up to 210 min. Representative microphotographs are shown (Figure 9a). Red dashed-line rectangles encompass the selected appositional and non-appositional zones that were analyzed (Figure 9b,c, respectively). After 30 min of pretreatment of the HeLa-Cx43EGFP cell line with Cyto-B, we observed a rapid and sustained increase in the number of fluorescent particles at the non-appositional plasma membrane, showing a 1.7 (60–90 min)-, 2 (120–150 min)- and 2.6-fold change (180–210 min) compared to the control (Figure 9e,f). However, in the appositional zone, we observed an initial increment (60–90 min) followed by a steady reduction in the arriving rate of Cx43EGFP fluorescent particles (Figure 9d,f). We also performed TIRF experiments on HeLa-Cx26GFP treated with Cyto-B. We found that trafficking to appositional and non-appositional plasma membranes was similar to that observed for Cx43EGFP HCs. The increment in the number of fluorescent particles containing Cx26GFP was observed after 180 min of treatment with Cyto-B in the non-appositional plasma membrane, presenting a 3.6-fold change versus control cells (Appendix A). In the appositional plasma membrane, the number of Cx26GFP fluorescent particles increased by 2.6-fold (120–150 min), followed by a reduction of fluorescent particles (Appendix A). Both experiments suggest a transient increase in the appositional plasma membrane followed by a reduction in the number of fluorescent particles. However, in the non-appositional plasma membrane, there is a constant increase in the number of fluorescent particles in HeLa-Cx43EGFP and HeLa-Cx26GFP.

### 2.5. Actin Depolymerization Produces Fast Changes in the Functional State of Cx43 GJCs and HCs

The reduction in dye coupling induced by actin depolymerization could be the consequence of (i) a reduced number of functional GJCs, (ii) a modification of channel permeability, or (iii) modified conductance properties. To evaluate possible rapid changes in the functional state of Cx43 GJCs induced by Cyto-B, we measure the electrical coupling using the two-whole cell voltage clamp technique [27,29]. Untreated cells exhibit large GJCs currents (Figure 10a, right panel) that are sensitive to 18 β-glycyrrhetinic acid (18β-GA), a GJCs inhibitor (Figure 10a, middle panel). Moreover, we observed a substantial time-dependent reduction in the amplitude of electrical coupling between pair cells after pretreatment with Cyto-B for 0.5, 1, and 2 h (Figure 10b). A 66.9% reduction in the electrical coupling was already observed at 30 min of incubation with Cyto-B, extending to 96% or more at two hours of treatment (Table 1). The generated transjunctional current (I_j_) after the application of the 200 ms voltages steps were plotted at the different applied voltages (Figure 10c).

On the other hand, for a more accurate determination of the effect of Cyto-B treatments on the functional state of Cx43 HCs, we performed electrophysiological recording using the whole cell voltage clamp technique [30,31]. We found that 10 min of treatment with Cyto-B is enough to increase HC currents in cells transfected with Cx43 (Figure 11b). Quantification analysis reveals that in the presence of Cyto-B, the evoked current increased rapidly and was abolished by La^3+^ (Figure 11d). Moreover, recordings obtained from HeLa Parental cells treated with Cyto-B and subsequent quantification show no changes in the current amplitude (Figure 11c,e). These data suggest that actin depolymerization also induced a rapid increment in the voltage-dependent ionic currents through Cx43 HCs.

## 3. Discussion

The present study showed that the actin cytoskeleton and RhoA signaling pathway dynamically modulate the traffic of HCs and GJCs plaque formation for Cx43 and Cx26. The role of RhoA and actin fibers in the traffic [7,8,9,32,33,34], and the function of connexins [21,35,36,37], have been centered on GJCs. Few studies focus on HCs [26,38,39,40,41]. Here, we showed that RhoA and actin modulate the traffic of HCs to the appositional and non-appositional plasma membrane in different ways. We found that the actin cytoskeleton favors the formation of GJCs over free HCs, therefore controlling the ratio between HCs and GJCs at the cell membrane. Consistently, actin depolymerization reduces Cx43 GJCs activity, associated with an increment of the plasma membrane permeability produced by more Cx43 HCs activity. However, the functional state of HCs and GJCs made of Cx26 is not affected by actin filament depolymerization. In addition, the Cx26 GJC and HC functional states do not correlate with gap junction plaque size or increased trafficking of HCs to the non-appositional plasma membrane, suggesting that GJCs and HCs activity control by actin microfilaments and RhoA depends on the Cx isoform. 

There are some contradictory reports about the role of RhoA in the formation of GJCs. For example, one report showed that a reduction in RhoA expression correlates with less expression of Cx26 and Cx43 in human colonic smooth muscle cells [42] or reduced connexin-mediated ATP release in lung epithelial cells [43]. In contrast, the inhibition of RhoA produced upregulation of Cx26, Cx30, and Cx43 in human nasal epithelial cells [44] and astrocytes [41]. Moreover, activation of RhoA/ROCK promotes an increase of Cx43 expression in the nephron [45], suggesting that the effect of RhoA on Cx expression may be cell-type specific.

As mentioned earlier, Cx43 is translated in the ER, packaged into vesicles in the Golgi apparatus, and delivered to the cortical plasma membrane [46]. In contrast, Cx26 delivering to the plasma membrane bypasses the Golgi apparatus, representing a faster transport mechanism [5,9]. Nonetheless, our findings indicate that the traffic of Cx43 or Cx26 to the plasma membrane is similarly affected by actin depolymerization, no matter whether it was induced by inhibiting RhoA or by Cyto-B treatment. Proper trafficking of connexins to the plasma membrane can avoid deleterious effects for the cell. For example, remodeling of Cx43 from the intercalated disc to a lateral location impairs cardiac function in mdx mice due to reduced GJCs formation [47]. Moreover, some authors reported that actin is part of a cytoskeleton-based forward trafficking system that mediates the specific delivery of Cx43 to the intercalated discs in cardiomyocytes to serve as GJCs [29]. The proper delivery of Cx43 to the intercalated discs seems to be controlled by the expression of a Cx43-internally translated minor C-terminus isoform (20 kDa), named GJA1-20k, generated by the translation initiation at an IRES site in Cx43 mRNA [48]. GJA1-20k might stabilize actin filaments for forwarding microtubule-dependent trafficking of Cx43 to intercalated discs [49]. However, there are no reports of this type of regulation for Cx26, although IRES exists in the 5’non-coding region of human Cx26 mRNA [50]. Therefore, we think that, at least in HeLa cells, the actin-dependent trafficking of Cx26 or Cx43 to appositional cell membranes is independent of the expression of regulatory subunits such as GJA1-20k. Finally, some reports suggest the role of actin microfilaments in the delivery of other ion channels to the plasma membrane. For example, in rat cardiomyocytes, the actin filaments’ stabilization induces an increase in the traffic of L-type Ca^2+^ channels from the perinuclear area to the T-tubule, where these channels take place in Ca^2+^ homeostasis regulation [51].

Our results show that Cx43 gap junction plaque size correlates well with GJCs function, which agrees with previous studies showing that clustering Cx43 GJCs into gap junction plaques is necessary for the active state GJCs [52]. However, this may not be a general property because the significant reduction in Cx26 gap junction plaque size induced by inhibition of RhoA or by treatment with Cyto-B did not affect the intercellular coupling. 

Small Rho GTPases are significant regulators of cellular junctions and the actin cytoskeleton. Our results partially agree with previous findings for Cx43, showing that RhoA is involved in regulating the functional state of GJCs in rat cardiac myocytes. Derangeon et al. observed that the specific inhibition of RhoA by the exoenzyme C3 decreases the size and reduces the permeability of GJCs formed by Cx43 in ventricular myocytes [21]. This correlates with our findings, in which the inhibition of RhoA with C3 reduces the number of actin stress fibers concomitant with a reduction in the plaque size index in GJs formed by Cx43, and we extended this observation to GJs plaques formed by Cx26. Consistent with the data provided by Derangeon et al., 2008 [21], we found impairments in the diffusion of LY between HeLa-Cx43 cells treated with C3 or with Cyto-B. Despite some similarities, our discovery that 30 min of treatment with Cyto-B induced a significant reduction in the electrical coupling suggests that the actin depolymerization produces a rapid and direct effect on the functional state of GJCs. This effect is Cx-type dependent because the Cx26 GJC functional state was not affected by RhoA inhibition or Cyto-B treatments.

Interestingly, we found that overexpression of RhoA had no effect on the number and size of Cx43 and Cx26 GJCs plaques (Appendix A). One plausible explanation is the potential saturation of mature gap junction plaques occupying the entire appositional plasma membrane surface available, which indeed depends on the cell-cell adhesion area. This area is initially regulated by the expression and localization of adhesion proteins such as cadherins and associated scaffolding proteins such as ZO-1 [53]. Therefore, even with increased RhoA activity, the extent of gap junction plaque formation may be the available adhesion surface area. In addition, evidence suggests that plaque formation occurs within specific membrane domains of the appositional membrane, which could restrict the regions where gap junction plaques form [54]. This spatial restriction might be independent of RhoA activity or the actin cytoskeleton dynamics, explaining why a fully formed gap junction plaque does not enlarge, even when HCs are available. 

On the other hand, we found that inhibition of RhoA or treatment with Cyto-B increases the rate of ethidium uptake in cells expressing Cx43 but not Cx26. It suggests that actin depolymerization increases HC activity by a Cx-type-dependent mechanism. Our finding was in line with previous work, in which the inhibition of Cx43 HCs mediated by thrombin was reverted by treating cardiomyocytes with C3 [55]. Additionally, we found that the HeLa-Cx43EGFP cell line treated with 4 µM Cyto-B increases the whole cell currents compared to the untreated or untransfected cells. The increase in Cx43 HCs activity determined by electrophysiology was faster and observed only 10 min after incubation with Cyto-B, which is earlier than the increment in HCs trafficking to non-appositional cell membranes observed in TIRF microscopy experiments, suggesting that actin depolymerization directly increases HCs activity. However, we cannot discard that the sustainable increase in HC trafficking contributes to the dye uptake (HC activity) observed after 60 min treatment with Cyto-B. Similarly, studies performed in human embryonic kidney (HEK) cells expressing Kv4.2, a potassium channel expressed in the brain and heart [56], show an increase in the whole cell currents after 1 h of treatment with Cyto-D. The authors found no changes in the voltage dependence of activation and inactivation of macroscopic currents. They found no changes in channel conductance, open probability, and kinetics at single-channel levels. They conclude that treatment with Cyto-D increases the numbers and alters the distribution of Kv4.2 channels at the plasma membrane [57]. Similar types of biophysics analyses are necessary to determine the mechanism of the actin polymerization/depolymerization over Cx43 HC and GJC functional states.

The effect of RhoA and actin cytoskeleton on the functional state of HCs and GJCs may also depend on specific cell types or conditions. For example, Kim et al. (2020) found that inhibition of RhoA reduces HCs and GJCs activity in a scrapie-infected neuronal cell line that was associated with increased interaction between RhoA and Cx43 [58]. In addition, an association between increments in RhoA and reduction of Cx43 phosphorylation in intercalated discs of cardiomyocytes was observed in aged rats presenting long-QT [59]. Therefore, we cannot discard that the effect induced by inhibition of RhoA or actin depolymerization on the functional state of HCs and GJCs can be related to changes in Cx43 phosphorylation.

The simultaneous visualization of the HCs trafficking to the appositional and non-appositional plasma membrane using TIRF microscopy showed that whereas actin depolymerization reduces the forward trafficking to appositional membranes, the transport of HCs to non-appositional membranes increased exponentially. This could be a general mechanism for Cxs, as the only difference observed between Cx43 and Cx26 is the temporal course. Both Cxs show a steady increase in the number of fluorescent particles in the non-appositional plasma membrane (Figure 9e,f; Appendix A) and a transient increase in the number of particles followed by a decrease in the number of particles at appositional cell membranes (Figure 9d,f;Appendix A). The observation of direct trafficking of HCs to appositional membranes for gap junction formation is consistent with previous findings in Cx43 [13,26]. However, the data reported by Shaw et al. suggest that Cx43 HCs are transported directly to appositional membranes by a microtubule-dependent pathway that includes the microtubule plus-end-tracking protein (EB1) and its interacting protein, p150 [13]. Here, we show that the direct transport of HCs to appositional membranes requires an intact actin cytoskeleton.

Although we did not observe major changes in cell morphology produced by the treatments, we cannot discard that part of the findings can be explained by the alteration of normal cell polarity produced by the changes in the dynamic of actin polymerization/depolymerization induced by the Cyto-B or the inhibition of RhoA. On the other hand, a possible role of connexin in the establishment of cell polarity has been proposed. For example, the absence of Cx43 disrupts the apicolateral distribution of ZO-1 in breast luminal epithelial cells causing alteration in cell polarity and leading to a random mitotic spindle orientation [60]. Moreover, cardiac defects observed in the Cx43 KO mice arise from the disruption of cell polarity, probably by disruption of Cx43-tubulin interactions [61].

Finally, the inhibition of RhoA or actin polymerization can directly affect protein complexes with a potential role in connexin trafficking, particularly protein complexes formed by cadherins and cell surface scaffolding proteins EB1/P150 and β-catenin located at appositional cell membranes, which have been described as major players in the directional forward trafficking of Cx43 to appositional membranes [13]. EB1 is a microtubule plus-end tracking protein that links tubulin cytoskeleton to N-cadherin in appositional cell membranes, which target Cx43-carrying vesicles through interactions with P150 Dynein/Dynactin/β-catenin protein complex to adherens junctions [13]. Likewise, N-cadherin promotes downstream RhoA and Rac1 activation, which is critical for determining the plasma membrane distribution of Cx43 in cardiac myocytes [62]. Recently, it was found that EB1 also binds filamentous actin [63], opening the possibility that perturbation in filamentous actin may affect EB1 localization, producing mistargeting of HC trafficking. On the other hand, it is known that filamentous actin can be associated with adherens junctions by direct interaction with α-catenin, which links cadherins through β-catenin and p120-catenin. This interaction results in active stabilization of the adherens junction [64]. In addition, RhoA stabilizes adherens junctions in migrating epithelial cells and regulates junction contractility, producing redistribution of appositional membrane proteins [65,66]. Therefore, it is possible that redistribution of HC trafficking to cell surface produced by RhoA inhibition or F-actin depolymerization can be a consequence of the instability of adherens junctions and redistribution of cadherins to non-appositional cell membranes.

Controlling the balance between the trafficking of HCs to non-appositional and appositional membranes by the actin cytoskeleton may have significant physiological consequences, for example, in cell migration and mitosis processes where the actin cytoskeleton is rapidly redistributed. Overall, our results suggest that although actin dynamics and RhoA critically regulate the Cx43 and Cx26 traffic to the plasma membrane, their role in regulating the functional state of HCs and GJCs depends on the Cx isoform.

## 4. Materials and Methods

### 4.1. Antibodies, Drugs, and Constructs

Cytochalasin B (Cyto-B) (Sigma-Aldrich, St. Louis, MO, USA) was used to inhibit actin polymerization at 37 °C. Exoenzyme C3 Transferase from *Clostridium botulinum* (C3, Cytoskeleton Inc, Denver, CO, USA) was used to inhibit the RhoA GTPase protein. Actin stress fibers visualization was, with phalloidin, conjugated with tetramethylrhodamine (Phalloidin-TRITC from Sigma-Aldrich, St. Louis, MO, USA). GJC activity was assessed by microinjection using Lucifer Yellow (LY) (Sigma-Aldrich, St. Louis, MO, USA). The HC activity was measured by addressing the ethidium bromide incorporation into cells (Sigma-Aldrich St. Louis, MO, USA). To block the functional state of GJCs and HCs, we used 18β-glycyrrhetinic acid (β-GA), carbenoxolone (CBX), and lanthanum (La^3+^) (Sigma-Aldrich, St. Louis, MO, USA). Constructs used in this study, RhoAWT, RhoAV14, RhoAN19, siRNARhoA, siRNAUnR, and the empty vector, were kindly provided by Dr. Stéphane Gasman [67], (INCI, Strasbourg, France).

### 4.2. Cell Culture

Untransfected HeLa Parental cells and stably transfected HeLa-Cx43, HeLa-Cx26, HeLa-Cx43-EGFP, and HeLa-Cx26-GFP were grown at 60–70% of confluence before experimental procedures. Cell lines were maintained in a tissue culture medium, DMEM (GIBCO, Invitrogen, Waltham, MA, USA) supplemented with 10% fetal bovine serum (Corning, New York, NY, USA), at 37 °C and 5% CO_2_ atmosphere.

### 4.3. Immunofluorescence, Actin Fibers Quantification, and Plaque Size Index

Immunofluorescence was performed as previously described [26]. Briefly, cells were fixed with 4% paraformaldehyde in phosphate-buffered saline (1X PBS; pH 7.4), permeabilized with 1% Triton X-100 in 1X PBS and blocked with 2% bovine serum albumin (BSA). Rabbit polyclonal primary antibodies against Cx43 and Cx26 were from (Invitrogen, Waltham, MA, USA), and the Cy2-conjugated goat anti-rabbit IgG secondary antibodies were from (Jackson InmunoResearch, Baltimore, PA, USA). When noted, incubation with 1 µM phalloidin-TRITC was applied to visualize actin fibers. Images were acquired using a Nikon Eclipse TE-2000U epifluorescence inverted microscope. The images were analyzed and quantified using the Nikon ACT-2U (Nikon, Tokyo, Japan) and the ImageJ (https://imagej.nih.gov/ij/, accessed on 1 March 2024) software, respectively. To quantify the numbers of actin fibers, all images were processed the same way using the subtract background tool in ImageJ. Afterward, a straight line was drawn in one of the cells in the field, and a plot profile was performed. The resulting plot profile was used to count the number of spikes of every analyzed cell under different conditions. The number of spikes represented the number of actin fibers. The gap junction size was evaluated through the “plaque size index”. The index was defined as the length of gap junctions within the contact zone of two adjacent cells, divided by the length of the contact zone of these cells: plaque size index (a.u.) = Length of GJs/Length of the contact zone. The composition of the final figures was achieved using Adobe Photoshop 2021 (Adobe Systems Inc, San Jose, CA, USA).

### 4.4. GJCs Activity

The functional state of GJCs was determined by measuring the diffusion of Lucifer Yellow (LY, MW 457, net charge -2, Sigma-Aldrich St. Louis, MO, USA), as previously described [68]. In brief, HeLa cells seeded in 12 mm glass coverslips were microinjected using an InjectMan-FemtoJet system (Eppendorf, Hamburg, Germany) coupled to microcapillaries (Femtotips II, Eppendorf) filled with a solution of 4% of LY. The magnitude of dye spreading was observed using a microscopy equipped for epifluorescence Nikon Eclipse TE-2000U (Nikon, Tokyo, Japan). The dye coupling index was calculated as the mean number of cells showing dye-positive staining.

### 4.5. Recording of GJCs Currents

Electrophysiological recordings in pairs of transfected HeLa cells in a cluster of cells in a non-confluent monolayer were performed as previously described [26,69]. Briefly, HeLa cells seeded in coverslips were placed in an experimental chamber containing Ham’s F12 media as the extracellular solution. Cells were observed using an inverted BX51WI Olympus microscope. Micropipettes were pulled from microcapillary borosilicate glass (World Precision Instruments, Sarasota, FL, USA using a micropipette puller (Sutter Instrument, Novato, CA, USA). The resistance of the resulting micropipettes was 7–10 megaOhms. The micropipettes were filled with intracellular solution containing (in mM): 10 KCl, 10 HEPES, 2-ethylene glycol tetraacetic acid (EGTA), 125 potassium gluconate, and 2-sodium adenosine 5’-triphosphate. Electrophysiological recordings in pairs of transfected HeLa cells were made by double whole-cell patch-clamp configuration using two patch clamp amplifiers (HEKA EPC7 Plus, Lambrecht, Germany; and Warner PC-501 A, Hamden, CT, USA). Cells were clamped at −60 mV, and a series of 200-ms voltage steps from −100 to +10 mV in increments of 10 mV were applied. The transjunctional current (I_j_) was recorded in the non-stimulated cell and plotted against the resulting voltage, which was calculated as follows: V_j_ = V_a_ − V_b_; where V_j_ is the resulting voltage, V_a_ is the voltage step of the protocol on the stimulated cell, and V_b_ is −60 mV (clamped voltage of the non-stimulated cell). Data was acquired with IGOR 6.0 software using an analog-digital converter card (BNC-2090; National Instruments, Austin, TX, USA). 

### 4.6. HCs Activity

Cells plated on glass coverslips were bathed in physiological extracellular solution containing 5 µM of ethidium bromide (ethidium, 314 Dalton/+1, Sigma-Aldrich St. Louis, MO, USA). Basal fluorescence intensity from the cells was recorded for 5 min using a 40× objective in a Nikon Eclipse TE-2000U inverted microscopy and subjected to dye uptake time-lapse imaging [26,27]. First, we assessed the ethidium uptake for 5 min in the presence of divalent ions (Ca^2+^/Mg^2+^) because, in these conditions, HCs are preferentially closed. After that, to increment the HC activity, the assay was performed in a medium without divalent cation (Divalent Cation-Free solution, DCSF) for 20 min because the divalent ions regulate the opening of HCs [70,71]. Finally, in the last 5 min of the experiment, we applied Lanthanum (La^3+^, 200 µM) to inhibit the HC activity. ImageJ software version 1.53k was used to perform analysis and quantification of fluorescence intensity (https://imagej.nih.gov/ij/; accessed on 1 March 2024). The microscope and camera settings (Nikon DS-2WBc fast-cooled monochromatic digital camera (8-bit) every 60 s) remained the same in all experiments.

### 4.7. Electrophysiological Recording of HCs

HeLa Parental or HeLa-Cx43 cell lines plated on 12 mm glass coverslips were placed in an inverted microscopy equipped with phase contrast and fluorescence system illumination. The chamber was perfused with the extracellular solution contained (in mM): 140 NaCl, 5.4 CsCl, 1 MgCl_2_, 1.8 CaCl_2_, 2 BaCl_2_, 10 HEPES, pH adjusted to 7.4, and 60 µM Mibefradil, a Cl^−^ channel blocker, applied to avoid interference with volume-regulated anion channels (VRAC) [72]. Isolated cells were recorded using a pipette solution containing (in mM): 130 CsCl, 10 AspNa, 0.26 CaCl2, 1 MgCl_2_, 3 MgATP, 3 EGTA, 7 TEACl, and 5 HEPES, with pH adjusted to 7.2. The pipette resistance was 5–10 megaOhms. The protocol to determine macroscopic currents consisted of a pulse of 6 s in length with increments of 30 mV from 0 to +90 mV. Patch pipettes connected to an amplifier (Multiclamp 700B, Axon CNS Molecular devices, San Jose, CA, USA) were controlled by a micromanipulator (MP-225; Sutter Instrument, Novato, CA, USA). Analog-digital data conversion was performed using an Axon Digidata 1550 Low Noise Data Acquisition system (Axon Instruments, Molecular Devices Electrophysiology, San Jose, CA, USA).

### 4.8. Total Internal Reflection Fluorescence (TIRF) Imaging

The TIRF technique is based on the formation of a thin electromagnetic area known as an evanescent wave, used to excite fluorophores located close to the cell membrane at a distance no greater than 100–200 nm between the membrane and the cover glass carrying the specimen, allowing the dynamic study of protein transport and its incorporation into the plasma membrane [73]. HeLa cells expressing Cx43-EGFP or Cx26-GFP were plated at 50–60% confluence on coverslips and observed using a Nikon (Eclipse TE-2000U) inverted microscope with a 100X Apo TIRF oil objective with a numerical aperture (NA) = 1.49, using a 488 nm laser. Image acquisition was in cells bathed in HAM F-12, applying a continuous rate of one frame every five minutes for a total of 30 min with 500 ms of exposure per frame. Afterward, Cyto-B was added to the medium, and images were acquired every five minutes for 30 min for 210 min between 30 min of non-recording periods to avoid excessive laser stimulation and photobleaching. Fluorescent particles in appositional zones (areas of cell-cell contact) and non-positional zones (away from the contact zones) were analyzed. All TIRF images were processed, analyzed, and quantified using ImageJ software (https://imagej.nih.gov/ij/; accessed on 1 March 2024).

### 4.9. Statistical Analysis

Statistical analysis was performed using Student’s *t*-test. Data are presented as the mean ± standard error (SEM). For graph design and statistical analysis, GraphPad Prism 8 was used (GraphPad Software, San Diego, CA, USA). A *p*-value of ≤0.05 was considered significant.

## Figures and Tables

**Figure 1 ijms-25-07246-f001:**
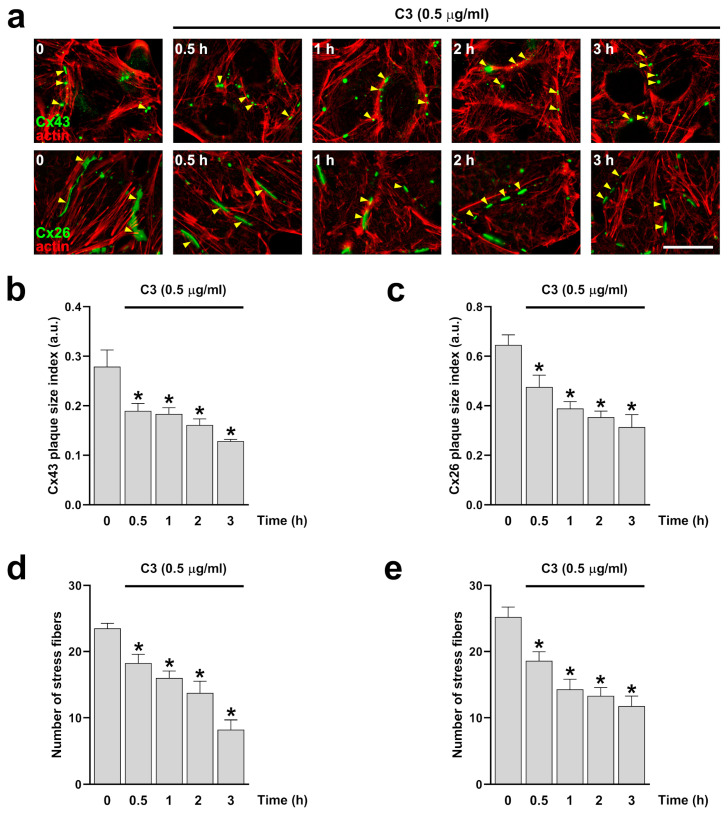
Inhibition of RhoA reduces the size of Cx26 and Cx43 GJCs plaques and the number of actin stress fibers. HeLa cells transfected with Cx43 ((**a**) upper panels) or Cx26 ((**a**) bottom panels) were exposed to C3 for 0.5, 1, 2, and 3 h. The effect of the drug treatment on plaque formation at the cell-cell borders (yellow arrowheads) was quantified on cells expressing Cx43 (**b**) or Cx26 (**c**). Quantification of actin fibers stained with phalloidin-TRITC was performed using ImageJ software. Data are presented as mean ± SEM (*n* = 5; 20 cell pairs per condition). The effect of the drug treatment on actin stress fiber formation was quantified on HeLa cells expressing Cx43 (**d**) and Cx26 (**e**). Data are presented as mean ± SEM (*n* = 5; at least 25 cells per condition); * *p* < 0.05. Scale bar: 7.2 µm.

**Figure 2 ijms-25-07246-f002:**
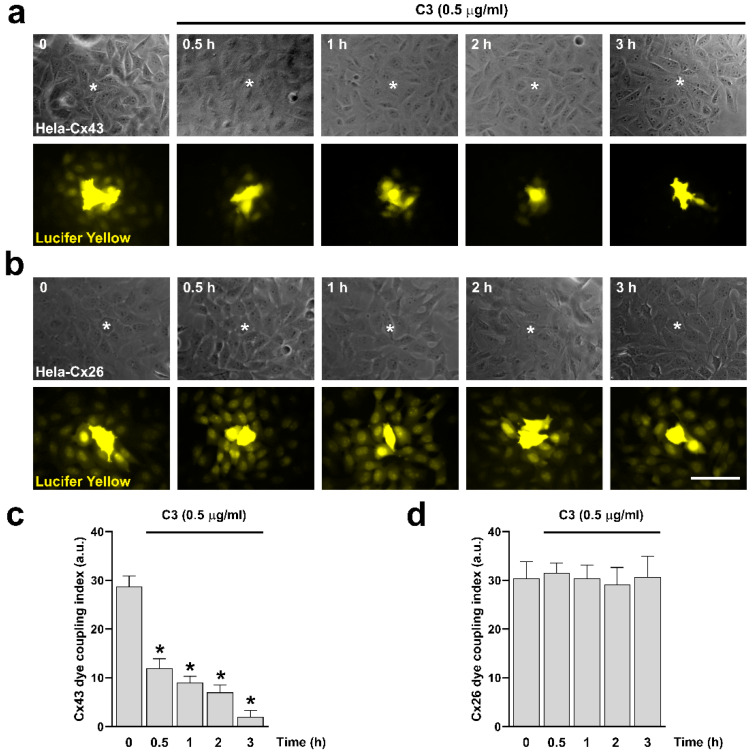
Inhibition of RhoA with C3 reduces Cx43 GJCs activity but does not affect the functional state of Cx26 GJCs. HeLa cells transfected with Cx43 (**a**) or Cx26 (**b**) were exposed to C3 for 0.5, 1, 2, and 3 h. Phase contrast ((**a**,**b**) upper panels) and diffusion of LY ((**a**,**b**) bottom panels) in HeLa-Cx43 and HeLa-Cx26 are shown. Graphs show the dye coupling index in HeLa cells transfected with Cx43 (**c**) and Cx26 (**d**). Data are presented as mean ± SEM (*n* = 5; 20 cells for each condition); * *p* < 0.05. Scale bar: 30 µm.

**Figure 3 ijms-25-07246-f003:**
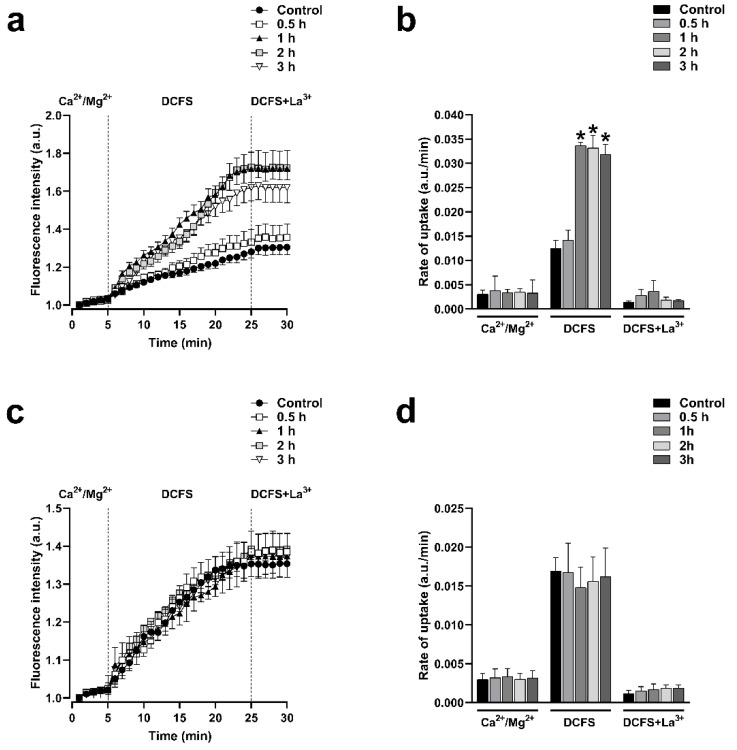
Inhibition of RhoA with C3 increases the activity of HCs formed by Cx43 but does not affect the activity of HCs formed by Cx26. Graph showing the time-lapse of ethidium uptake in HeLa-Cx43 treated with C3 for 0, 0.5, 1, or 3 h (**a**). Ethidium uptake rates were determined by calculating the uptake slope in cells expressing HeLa-Cx43 bathed with physiological (Ca^2+^/Mg^2+^), Ca^2+^/Mg^2+^-free (DCFS), and DCSF+La^3+^ solutions (**b**). Graph showing the time-lapse of ethidium uptake in HeLa-Cx26 treated with C3 for 0, 0.5, 1, or 3 h (**c**). Ethidium uptake rates were determined by calculating the uptake slope in cells expressing HeLa-Cx26 bathed with physiological (Ca^2+^/Mg^2+^), Ca^2+^/Mg^2+^-free (DCFS), and DCSF+La^3+^ solutions (**d**). Data are presented as mean ± SEM (*n* = 4); * *p* < 0.05.

**Figure 4 ijms-25-07246-f004:**
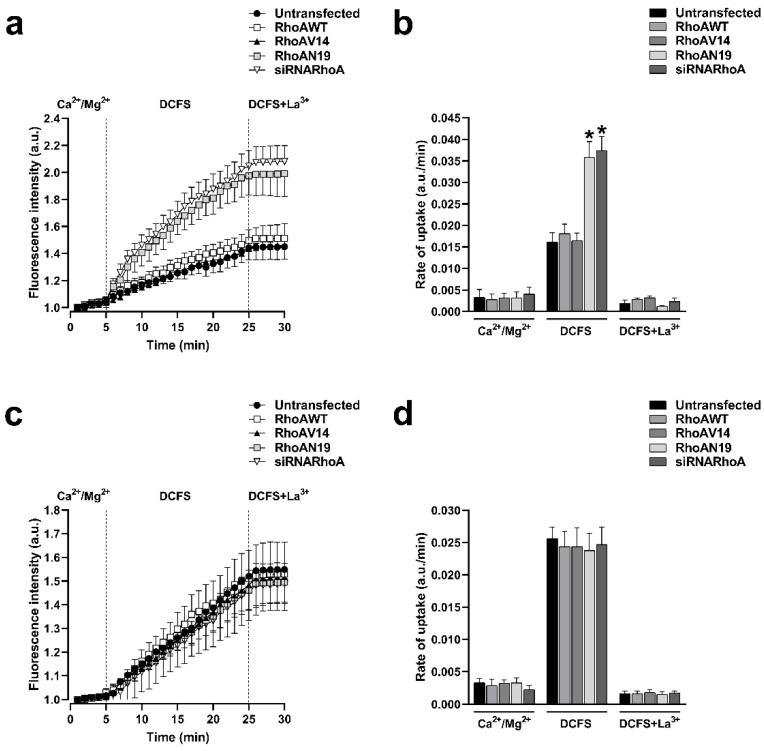
Inhibition of RhoA synthesis with siRNAs or expression of a dominant negative form of RhoA increases the activity of HCs formed by Cx43 but does not affect the activity of HCs formed by Cx26. Graph showing the time-lapse of ethidium uptake in HeLa-Cx43 transfected with RhoAWT, RhoAV14, RhoAN19, or siRNARhoA constructs (**a**). Ethidium uptake rates were determined by calculating the uptake slope in cells expressing HeLa-Cx43 bathed with physiological (Ca^2+^/Mg^2+^), Ca^2+^/Mg^2+^-free (DCFS), and DCSF+La^3+^ solutions (**b**). Graph showing the time-lapse of ethidium uptake in HeLa-Cx26 transfected with RhoAWT, RhoAV14, RhoAN19, or siRNARhoA constructs (**c**). Ethidium uptake rates were determined by calculating the uptake slope in cells expressing HeLa-Cx26 bathed with physiological (Ca^2+^/Mg^2+^), Ca^2+^/Mg^2+^-free (DCFS), and DCSF+La^3+^ solutions (**d**). Data are presented as mean ± SEM (*n* = 4); * *p* < 0.05.

**Figure 5 ijms-25-07246-f005:**
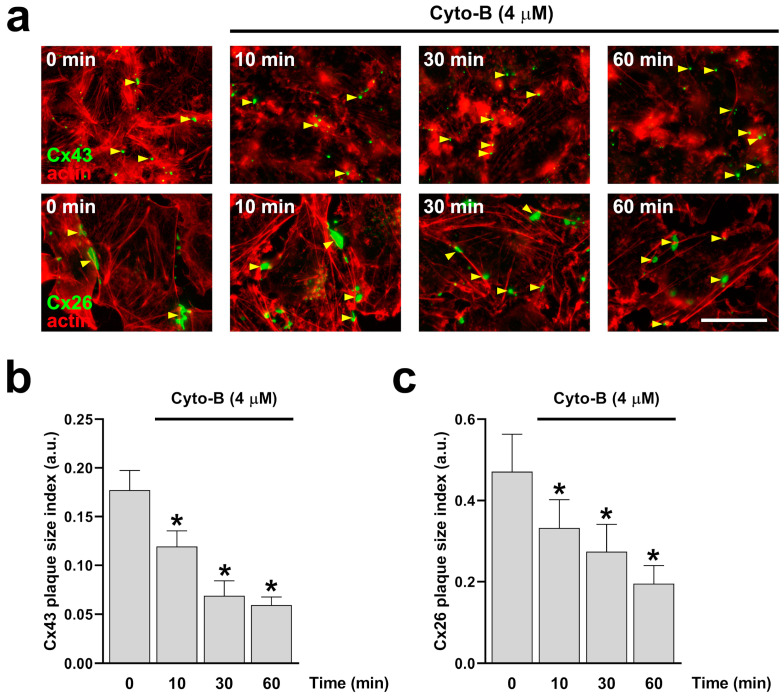
Time-dependent actin disruption with Cyto-B reduces the size of Cx43 and Cx26 GJCs plaque. HeLa cells transfected with Cx43 or Cx26 ((**a**) bottom panels) were exposed to Cyto-B for 10, 30, and 60 min. The effect of the drug treatment on plaque formation at the cell-cell borders (yellow arrowheads) was quantified on cells expressing Cx43 (**b**) or Cx26 (**c**). Data presented as mean ± SEM (*n* = 3; 15 cell pairs per condition in cells expressing Cx43; *n* = 3; 20 cell pairs per condition in cells expressing Cx26); * *p* < 0.05. Scale bar: 7.2 µm.

**Figure 6 ijms-25-07246-f006:**
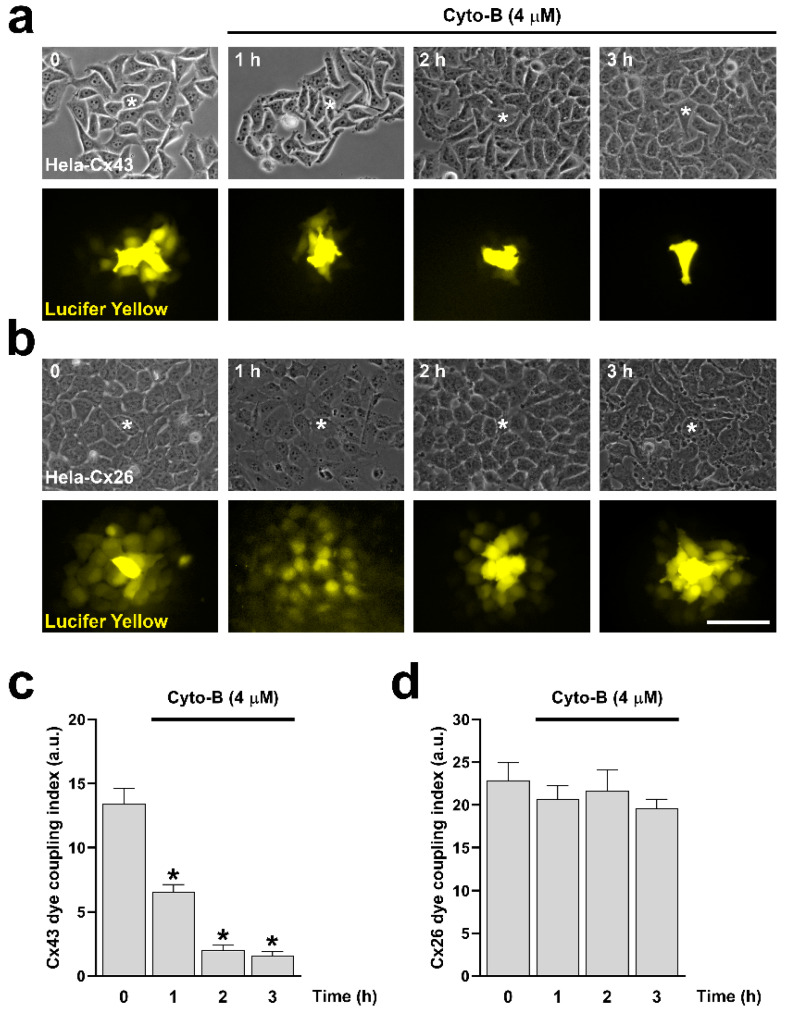
Cyto-B reduces Cx43 GJCs activity but does not affect the functional state of Cx26 GJCs. HeLa cells transfected with Cx43 (**a**) or Cx26 (**b**) were exposed to Cyto-B for 0, 1, 2, and 3 h. Phase contrast ((**a**,**b**) upper panels) and diffusion of LY ((**a**,**b**) bottom panels) of HeLa-Cx43 and HeLa-Cx26 cell lines are shown. The graph shows the dye coupling index in HeLa cells transfected with Cx43 (**c**) or Cx26 (**d**). Data are presented as mean ± SEM (*n* = 3; 5 impaled cells for each condition); * *p* < 0.05. Scale bar: 30 µm.

**Figure 7 ijms-25-07246-f007:**
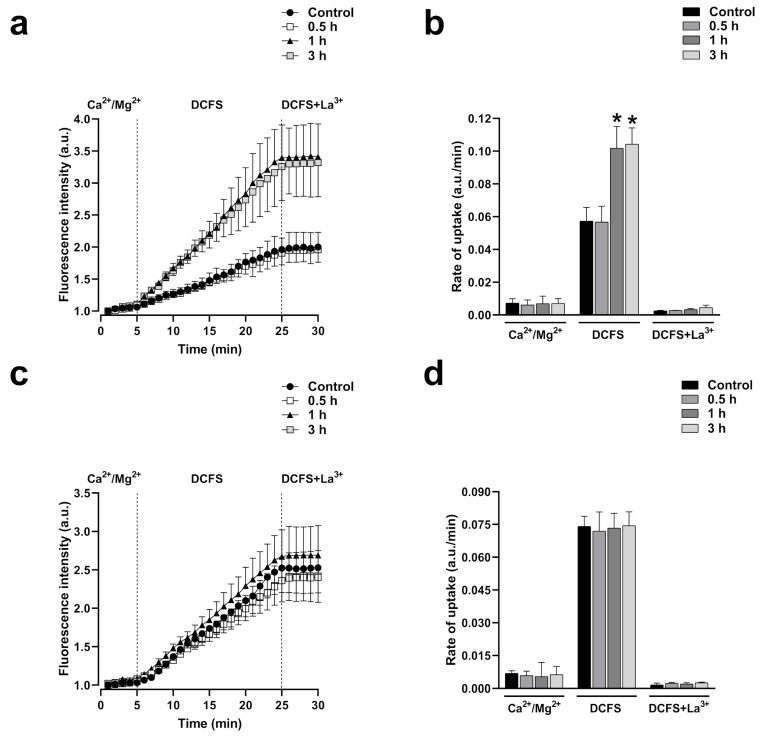
Cyto-B increases the activity of HCs formed by Cx43 but does not affect the activity of HCs formed by Cx26. Graph showing the time-lapse of ethidium uptake in HeLa-Cx43 exposed to Cyto-B for 0, 0.5, 1, or 3 h (**a**). Ethidium uptake rates were determined by calculating the uptake slope in cells expressing HeLa-Cx43 bathed with physiological (Ca^2+^/Mg^2+^), Ca^2+^/Mg^2+^-free (DCFS), and DCSF+La^3+^ solutions (**b**). Graph showing the time-lapse of ethidium uptake in HeLa-Cx26 exposed to Cyto-B for 0, 0.5, 1, or 3 h (**c**). Ethidium uptake rates were determined by calculating the uptake slope in cells expressing HeLa-Cx43 bathed with physiological (Ca^2+^/Mg^2+^), Ca^2+^/Mg^2+^-free (DCFS), and DCSF+La^3+^ solutions (**d**). Data are presented as mean ± SEM (*n* = 4); * *p* < 0.05.

**Figure 8 ijms-25-07246-f008:**
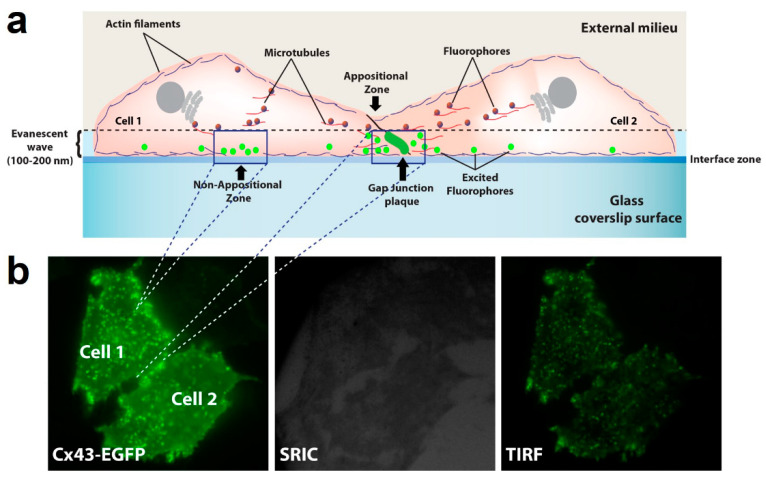
TIRFM configuration to study HCs transport to appositional and non-appositional plasma membrane zones. A representative cartoon showing two cells in juxtaposition and all the components involved in the data acquisition. TIRF microscopy configuration utilizes a critical angle, which creates a 100–200 nm-deep evanescent wave between the glass coverslip surface and cells to excite fluorophores located in the vicinity of the plasma membrane. Analysis was performed in the appositional and non-appositional zones (**a**). An example of how TIFR configuration is achieved under the microscopy is shown (**b**). Tagged connexins were illuminated in epifluorescence (left panel). Then, the surface reflective interference contrast (SRIC) filter ensures an appropriate field with cells attached to the glass (middle panel). Finally, it is switched to TIRF microscopy configuration for the data acquisition (right panel) (**b**).

**Figure 9 ijms-25-07246-f009:**
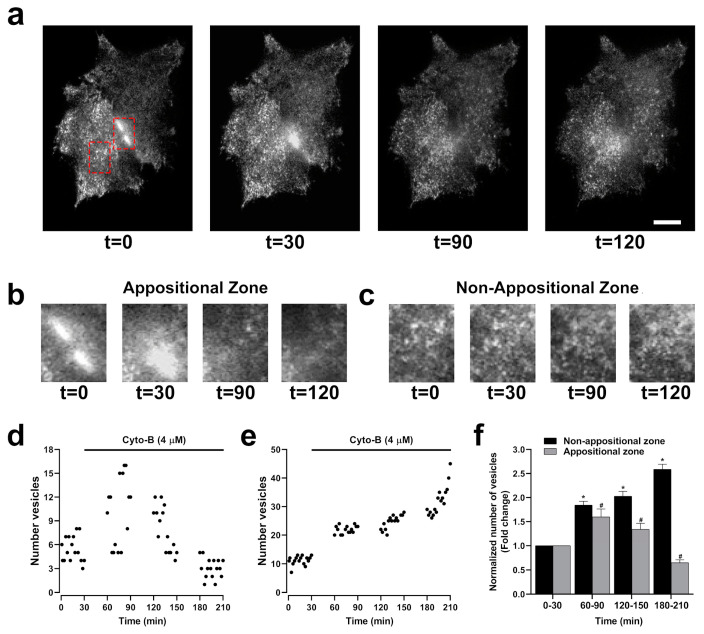
Cyto-B increases the number of vesicles in the non-appositional zone. Microphotographs of two cells in TIRF configuration show a time-lapse after Cyto-B (4 µM) treatment for up to 210 min. Appositional and non-appositional zones are encompassed by a red rectangle (**a**). Zoom-in of the red rectangle border appositional (**b**) and non-appositional zone of the cells (**c**). Representative graphs show the quantification of the number of vesicles in the appositional zone (**d**) and the non-appositional zone (**e**). The graph shows the fold change in the number of vesicles in the appositional and the non-appositional zone (**f**). Non-appositional zone (black bars) or appositional zone (gray bars) time intervals (60–90, 120–150, and 180–210 min) were compared to the control condition (0–30 min). Data are presented as mean ± SEM (*n* = 3) * *p* < 0.05. Scale bar: 5 µm.

**Figure 10 ijms-25-07246-f010:**
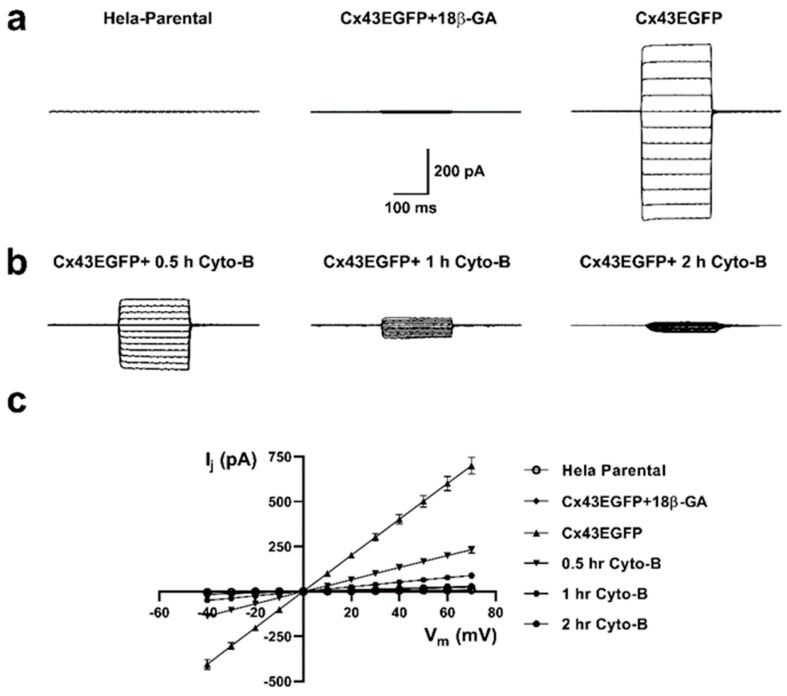
Cyto-B reduces the electrical coupling between Cx43EGFP pair cells. Voltage clamp steps were applied to voltages between −100 and +10 mV in 10-mV increments from a holding potential of −60 mV. Representative coupling traces of HeLa Parental (left panel), Cx43EGFP+18β-GA (middle panel), and Cx43EGFP (right panel) cell pairs recorded by double whole-cell patch clamp are shown (**a**). Reduction of the electrical coupling after treatment with Cyto-B for 0.5 h (left panel), 1 h (middle panel), and 2 h (right panel) are shown (**b**). Current-voltage relationship for pair cells of HeLa Parental and the evaluated conditions of Cx43EGFP. The data represents mean ± SEM (**c**). The number of experiments (*n*) and *p*-values are indicated in Table 1.

**Figure 11 ijms-25-07246-f011:**
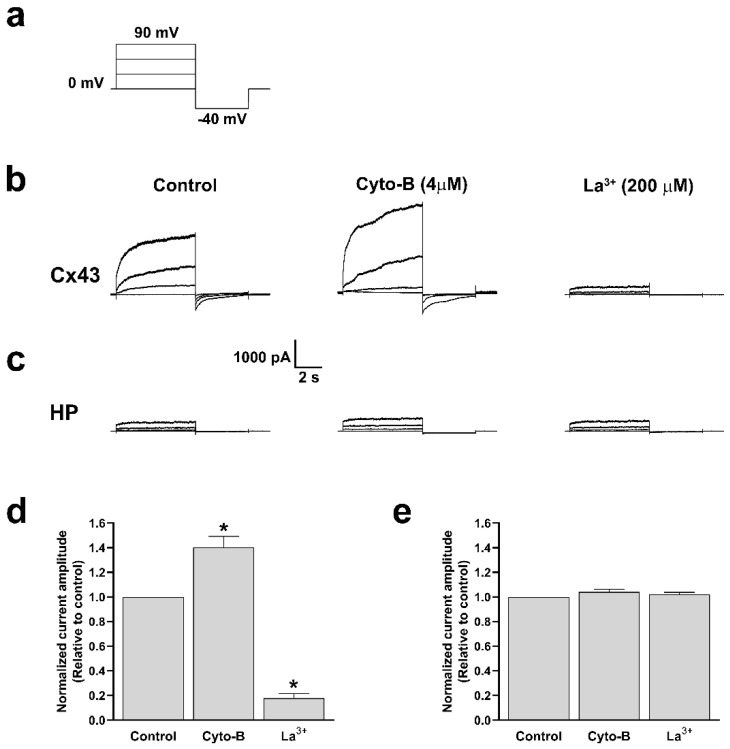
Cyto-B increases the ionic current through Cx43 HCs. Representation of voltage steps protocol used to evaluate total currents in HeLa-Cx43 or HeLa (HP) cell lines in whole-cell configuration. Cells were held at 0 mV and voltage steps (+30 mV increment, 6 s duration) were applied to promote HCs currents, followed by a −40 mV voltage step of 4 s length (**a**). Representative traces of Cx43 HCs or HeLa Parental currents in untreated control condition; 10 min in the presence of Cyto-B (4 µM) and La^3+^ (200 µM) (**b**,**c**). Graphs show the normalized amplitude current of HeLa-Cx43 (**d**) and HeLa (HP) (**e**). All treatments were compared to the control condition. Data are presented as mean ± SEM (*n* = 3); (* *p* < 0.05).

**Table 1 ijms-25-07246-t001:** Transjunctional current (I_j_) at +70 mV and *p*-values of each evaluated condition.

Condition	*n*	I_j_ ± S.E.M (pA)	*p*-Value
HeLa Parental	9	9.2 ± 3.5	3.79 × 10^−7^
Cx43EGFP+18β-GA	6	24.5 ± 6.9	3.6 × 10^−7^
Cx43EGFP	9	699.1 ± 46.5	n.a. ^1^
0.5-h Cyto-B	8	231 ± 19	2.03 × 10^−6^
1-h Cyto-B	6	88.4 ± 17.4	2.12 × 10^−7^
2-h Cyto-B	6	27.9 ± 2.9	4.84 × 10^−7^

^1^ Not determined. All the conditions were compared to Cx43EGFP.

## Data Availability

All data generated or analyzed during this study are included in this published article.

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
