# Peer review of "Differential Regulation of Hemichannels and Gap Junction Channels by RhoA GTPase and Actin Cytoskeleton: A Comparative Analysis of Cx43 and Cx26"

_ijms, 2024, doi:10.3390/ijms25137246_

Round 1

Reviewer 1 Report

Comments and Suggestions for Authors

This study by Jara et al. provides valuable insights into the role of RhoA and the actin cytoskeleton in the regulation of gap junction channels (GJCs) and hemichannels (HCs) composed of different connexin (Cx) proteins. The experiments demonstrate how RhoA inhibition and actin depolymerization affect the trafficking and activity of these channels. However, several points require further clarification and additional controls to strengthen the conclusions drawn.

  1. RhoA Overexpression and Connexin Plaques (Supplementary Fig.1):
    • Issue: The manuscript reports that overexpression or gain-of-function of RhoA does not affect connexin plaques, which is counterintuitive given that RhoA is known to influence actin dynamics and could be expected to enhance plaque formation.
    • Suggestion: The authors should provide a detailed explanation or hypothesis as to why RhoA overexpression does not result in larger or more numerous connexin plaques. It would be beneficial to include additional experiments or references to support this finding. Potential reasons such as compensatory mechanisms or saturation effects at high RhoA activity levels should be discussed.
  2. Diffusion of Lucifer Yellow (Fig.2):
    • Issue: The study attributes the diffusion of lucifer yellow to connexin proteins, but it is essential to confirm that this diffusion is not due to other channels or transport mechanisms.
    • Suggestion: To validate this conclusion, the authors should perform control experiments using connexin KO or employ specific blockers for connexin channels. These controls will ensure that the observed dye diffusion is indeed mediated by the targeted connexin proteins.
  3. Negative Controls for Dye Uptake (Fig.3 and Fig.4):
    • Issue: The current experiments lack negative controls, which are crucial to demonstrating that dye uptake is specifically due to the transfected connexin proteins.
    • Suggestion: Include experiments with wt HeLa cells that have not been transfected with connexin proteins. This will help determine if the dye uptake is connexin-specific or if other endogenous channels or transporters are involved. Such controls are necessary to unequivocally attribute the dye uptake to the connexin proteins being studied.
  4. Electrophysiological Recordings (Fig.11):
    • Issue: The electrophysiological data in Fig.11 is not entirely convincing. In panel b, there are fewer current traces after Cyto-B treatment compared to the control group, suggesting a lower open probability of the channels post-treatment. This observation contradicts the authors’ hypothesis. Additionally, there is a discrepancy between the current levels in Fig.11c for HeLa parental cells and those in Fig.10a.
    • Suggestion: The authors should identify conditions that allow for a full opening of the channels, ensuring consistent comparisons across different experimental conditions. This approach will help in accurately evaluating the effects of Cyto-B treatment. The authors should also address the discrepancy in the current levels observed in the HeLa parental cells between Fig.11c and Fig.10a, providing a plausible explanation or additional data to clarify this inconsistency.

Overall, while the study offers important findings on the role of RhoA and the actin cytoskeleton in connexin channel regulation, addressing these points will enhance the robustness and credibility of the conclusions. Implementing the suggested controls and providing additional explanations will significantly strengthen the manuscript.

Author Response

This study by Jara et al. provides valuable insights into the role of RhoA and the actin cytoskeleton in the regulation of gap junction channels (GJCs) and hemichannels (HCs) composed of different connexin (Cx) proteins. The experiments demonstrate how RhoA inhibition and actin depolymerization affect the trafficking and activity of these channels. However, several points require further clarification and additional controls to strengthen the conclusions drawn.

Comment 1.- RhoA Overexpression and Connexin Plaques (Supplementary Fig.1):

Issue: The manuscript reports that overexpression or gain-of-function of RhoA does not affect connexin plaques, which is counterintuitive given that RhoA is known to influence actin dynamics and could be expected to enhance plaque formation.

Suggestion: The authors should provide a detailed explanation or hypothesis as to why RhoA overexpression does not result in larger or more numerous connexin plaques. It would be beneficial to include additional experiments or references to support this finding. Potential reasons such as compensatory mechanisms or saturation effects at high RhoA activity levels should be discussed.

Response: Thank you for the valuable feedback. The most plausible explanation for the lack of effect observed with RhoA overexpression or gain-of-function on connexin plaques is the potential saturation of mature gap junction (GJ) plaques, occupying the entire appositional plasma membrane surface available for this purpose.  The total surface area available for GJ formation primarily depends on the cell-cell adhesion area. This area is initially regulated by the expression and localization of adhesion proteins such as Cadherins and associated scaffolding proteins like ZO-1 (Palatinus et al., 2011REF ). Therefore, even with increased RhoA activity, the extent of GJ plaque formation may be limited by the available adhesion surface area.  In addition, evidence suggests that plaque formation occurs within specific membrane domains of the appositional membrane, which could restrict the regions where GJ plaques form (Johnson et al., 2012 ). This spatial restriction might be independent of RhoA activity or actin cytoskeleton dynamics, explaining why a fully formed GJ plaque does not enlarge even when hemichannels (HCs) are available. Therefore, we include a paragraph in discussion to point it out.

Palatinus JA, O'Quinn MP, Barker RJ, Harris BS, Jourdan J, Gourdie RG. ZO-1 determines adherens and gap junction localization at intercalated disks. Am J Physiol Heart Circ Physiol. 2011 Feb;300(2):H583-94. doi: 10.1152/ajpheart.00999.2010. Epub 2010 Dec 3. PMID: 21131473; PMCID: PMC3044061. 

Johnson RG, Reynhout JK, TenBroek EM, Quade BJ, Yasumura T, Davidson KG, Sheridan JD, Rash JE. Gap junction assembly: roles for the formation plaque and regulation by the C-terminus of connexin43. Mol Biol Cell. 2012 Jan;23(1):71-86. doi: 10.1091/mbc.E11-02-0141. Epub 2011 Nov 2. PMID: 22049024; PMCID: PMC3248906.

Comment 2.- Diffusion of Lucifer Yellow (Fig.2):

Issue: The study attributes the diffusion of lucifer yellow to connexin proteins, but it is essential to confirm that this diffusion is not due to other channels or transport mechanisms.

Suggestion: To validate this conclusion, the authors should perform control experiments using connexin KO or employ specific blockers for connexin channels. These controls will ensure that the observed dye diffusion is indeed mediated by the targeted connexin proteins.

Response: We appreciate the reviewer's suggestion to incorporate the relevant controls. We have included the corresponding control data in a new supplementary figure (Supplementary Figure 2a, b). It is important to note that while we routinely perform these types of controls, we initially did not include them in this article because they had been documented in our previous publications. The HeLa parental cells used in our experiments do not exhibit positive dye coupling, as evidenced by the lack of Lucifer yellow or Calcein diffusion when using microinjection or FRAP techniques, respectively (Jara et al., 2012; Garcia et al., 2015; Abbott et al., 2021). Furthermore, we did not detect electrical coupling between HeLa parental cells (Jara et al., 2012; Garcia et al., 2015). Consistently, in cells transfected with Cx26 or Cx43, with or without fluorescent tags, dye coupling, and electrical coupling were significantly reduced when treated with Carbenoxolone or 18-β-glycyrrhetinic acid (Jara et al., 2012; Garcia et al., 2015; Abbott et al., 2021).

Comment 3.- Negative Controls for Dye Uptake (Fig.3 and Fig.4):

Issue: The current experiments lack negative controls, which are crucial to demonstrating that dye uptake is specifically due to the transfected connexin proteins.

Suggestion: Include experiments with wt HeLa cells that have not been transfected with connexin proteins. This will help determine if the dye uptake is connexin-specific or if other endogenous channels or transporters are involved. Such controls are necessary to unequivocally attribute the dye uptake to the connexin proteins being studied.

Response:  We thank the reviewer for the suggestion. As with our previous responses, we regularly conduct controls on dye uptake in HeLa untransfected cells and HeLa cells transfected with Cx26 or Cx43, and these controls have been published in our earlier articles (Jara et al., 2012; Garcia et al., 2015; Abbott et al., 2021). HeLa parental cells do not show significant dye uptake under normal conditions, nor does they increase dye uptake in the presence of divalent cation-free solutions, which induce hemichannel opening. Additionally, HeLa cells transfected with Cx26 or Cx43 show basal dye uptake that significantly increases in the presence of cation-free solutions. This uptake is markedly reduced by extracellular Lanthanum (La3+), a known hemichannel inhibitor (Jara et al., 2012; Garcia et al., 2015; Abbott et al., 2021).

As suggested by the reviewer, we have included a new supplementary figure (Supplementary Figure 2c) demonstrating these findings for HeLa parental cells and HeLa cells transfected with Cx43 and treated with Cytochalasin-B (Cyto-B). Consistent with our previous results, HeLa parental cells do not exhibit significant dye uptake, while HeLa cells expressing Cx43 show a significant increase in dye uptake when incubated in divalent cation-free solutions. This uptake is inhibited by extracellular Lanthanum. The further increase in uptake observed in HeLaCx43 cells treated with Cyto-B is greatly reduced by Lanthanum, supporting our interpretation that Cyto-B increases the number of functional hemichannels (Supplementary Figure 2c).

We do not believe it is necessary to show the same experiments for HeLa cells expressing Cx26, as Cyto-B did not increase dye uptake in these cells.

Comment 4.- Electrophysiological Recordings (Fig.11):

Issue: The electrophysiological data in Fig.11 is not entirely convincing. In panel b, there are fewer current traces after Cyto-B treatment compared to the control group, suggesting a lower open probability of the channels post-treatment. This observation contradicts the authors’ hypothesis. Additionally, there is a discrepancy between the current levels in Fig.11c for HeLa parental cells and those in Fig.10a.

Suggestion: The authors should identify conditions that allow for a full opening of the channels, ensuring consistent comparisons across different experimental conditions. This approach will help in accurately evaluating the effects of Cyto-B treatment. The authors should also address the discrepancy in the current levels observed in the HeLa parental cells between Fig.11c and Fig.10a, providing a plausible explanation or additional data to clarify this inconsistency.

Response: Regarding this question, we did not fully understand what the reviewer meant by “lower current traces after Cyto-B," and we believe there might be a misinterpretation. Figure 11b shows representative current traces following voltage clamp in HeLa Cx43 cells under control conditions and after a 10-minute treatment with Cyto-B. The amplitude of the current traces after voltage steps of +30, +60, and +90 mV are clearly larger in HeLa Cx43 cells treated with Cyto-B compared to the control condition, and these currents are also inhibited by lanthanum. The normalized quantification representing the change in hemichannel current amplitude is shown in Figure 11d.

 Regarding the following question: “Additionally, there is a discrepancy between the current levels in Figure 11c for HeLa parental cells and those in Figure 10a,” we believe the reviewer might be confused. Figure 10 shows current traces of gap junction channels, whereas Figure 11 shows current traces of hemichannels. The configurations of these experiments are very different: Figure 10 utilizes a dual whole-cell voltage clamp between two adjacent cells, while Figure 11 uses a whole-cell voltage clamp in a single isolated cell, which also shows some capacitive currents.

Reviewer 2 Report

Comments and Suggestions for Authors
  1. Some grammars and writings should be revised and polished. Here are some examples in the abstract

Ex. line 16, . Previous researches support….….

Ex. line 27, These results support...

Ex. line 31, communications….….

Ex. line 32, whereby the activity of Cx43 HCs and GJCs are differentially affected, but Cx26 channels remain unchanged.

Ex. line 256, TIRF configuration..

1.      Figure 1, 2, 5, 6. According to the legends, n=20 to 25 cells for statistics. So, were the cells grown in the same well? Theoretically, n=20 cells in the same well represent n=1 with 20 repeats (counts).

2.      Figure 8, time-lapse

In Figure 9, line 256, it should be TIRF configuration.

Comments on the Quality of English Language

Minor editing of English language required

Author Response

Some grammars and writings should be revised and polished. Here are some examples in the abstract

Ex. line 16, . Previous researches support….….

Ex. line 27, These results support...

Ex. line 31, communications….….

Ex. line 32, whereby the activity of Cx43 HCs and GJCs are differentially affected, but Cx26 channels remain unchanged.

Ex. line 256, TIRF configuration..

  1. Figure 1, 2, 5, 6. According to the legends, n=20 to 25 cells for statistics. So, were the cells grown in the same well? Theoretically, n=20 cells in the same well represent n=1 with 20 repeats (counts).
  2. Figure 8, time-lapse

In Figure 9, line 256, it should be TIRF configuration.

Response: We thank to reviewer 2 for their comments and suggestion to improve the article.  We have done a revision through all the text to improve the grammar and writing defects noticed by the reviewer